# Piceatannol Inhibits the Immunostimulatory Functions of Dendritic Cells and Alleviates Experimental Arthritis

**DOI:** 10.3390/ijms26083626

**Published:** 2025-04-11

**Authors:** Luyang Han, Peng Han, Yanbo Zhu, Jiawei Dong, Zhenyang Guan, Yuekang Xu, Jinyao Li, Xiaoying Liu

**Affiliations:** Xinjiang Key Laboratory of Biological Resources and Genetic Engineering, College of Life Science and Technology, Xinjiang University, Urumqi 830017, China; 107552201042@stu.xju.edu.cn (L.H.); hpd@stu.xju.edu.cn (P.H.); 107552301115@stu.xju.edu.cn (Y.Z.); 107552303692@stu.xju.edu.cn (J.D.); 107552301086@stu.xju.edu.cn (Z.G.); yuekang.xu@hotmail.com (Y.X.)

**Keywords:** piceatannol, dendritic cells, rheumatoid arthritis

## Abstract

Rheumatoid arthritis (RA) is a highly prevalent systemic autoimmune disease. Recently, natural small molecules have been explored as alternative therapeutic agents. *Iris halophila Pall* is the traditional herbal medicine, and it is rich in active ingredients with anti-inflammatory and immunomodulatory effects. In our previous study, LC-MS analysis revealed that piceatannol (PIC) is one of the primary active ingredients in the root of Iris tectorum. The purpose of this study was to explore the immunomodulatory effects of PIC on the maturation and function of dendritic cells, as well as on experimental arthritis induced by complete Freund’s adjuvant (CFA) and incomplete Freund’s adjuvant (IFA). Additionally, we aimed to probe into the potential mechanisms underlying the effects of PIC. We first verified the immunosuppressive effect of PIC using flow cytometry and an ELISA. The immunosuppressive mechanism of PIC on dendritic cells (DCs) was investigated through a joint analysis of network pharmacology and Western blotting. Our findings revealed that under Lipopolysaccharide (LPS)-induced inflammatory conditions, PIC could restrain the maturation and function of DCs (*p* < 0.001) and decrease the secretion of inflammatory cytokines (*p* < 0.001) compared to the LPS group. Furthermore, PIC suppressed the activation and polarization of CD4^+^ T cells, resulting in a decreased proportion of Th1 and Th17 cells (*p* < 0.001), ultimately improving the symptoms of CFA-induced arthritis in comparison to the model group. The PIC-induced shift in the T helper cell differentiation correlated with the secretion of polarizing cytokines from DCs in the AIA model. Mechanistically, PIC exerted its immunosuppressive function mainly by down-regulating the Mitogen-Activated Protein Kinase (MAPK) and Nuclear Factor kappa-B (NF-κB) signaling pathways. Collectively, these data unveil the anti-inflammatory mechanisms of a traditional medicine via the inhibition of the immune activation function of DCs in vivo and open up a therapeutic approach for autoinflammatory diseases.

## 1. Introduction

Rheumatoid arthritis (RA) is a common chronic systemic autoimmune disease, with a global morbidity of about 1% [1]. Its main features include the proliferation of synovial tissue, the formation of panni [2], the infiltration of inflammatory cells in the interstitium, and the destruction of cartilage and joint tissue [3]. Furthermore, RA can also lead to significant complications such as osteoporosis, ocular inflammation, and cardiovascular diseases [4]. Currently, various treatments for RA are widely utilized, including glucocorticoids (Dexamethasone and Prednisone) [5], NSAIDs (Aspirin and Celecoxib) [6], DMARDs (Iguratimod and Hydroxychloroquine) [7], as well as novel immunosuppressants (Methotrexate and Leflunomide) [8]. Although these drugs can alleviate inflammation and relieve joint pain, they do not provide a complete cure, and long-term use may result in serious side effects. Therefore, the development of effective therapeutic drugs that are low in toxicity, multi-targeted, and of natural origin remains a top priority.

The nosogenesis of RA is complex, primarily associated with an imbalance of immune cell subgroups, immune dysfunction, and the excessive activation of immune cells [9]. This complexity is largely attributed to chronic synovial inflammation caused by the deposition of synovial immune complexes and the infiltration of inflammatory cells [10]. Immune cells such as dendritic cells (DCs), fibroblast-like synoviocytes, macrophages, T and B lymphocytes, and neutrophils are abnormally activated in the joints of RA patients [11,12]. Among them, the T cell-mediated immunoinflammatory response is the main cause of RA pathogenesis. Under the influence of chemokines, activated T cells migrate from peripheral blood to the joints, where they further differentiate into effector T cells. This process disrupts the normal immune balance, leading to joint synovial inflammation, hyperplasia, and damage to or the deformity of cartilage and bone [13].

Pathogenic T cell-mediated joint destruction requires the activation of naïve T cells by professional antigen-presenting cells (APCs), followed by their differentiation into effector T cells under the stimulation of polarizing cytokines secreted by these APCs. Among these, DCs are the only APCs capable of effectively activating naïve T cells in vivo [14]. During maturation, MHC I/II and co-stimulatory molecules are highly expressed on the surfaces of DCs, which exhibit unique abilities to regulate the inflammatory response in the inflammatory synovial tissue of RA [15,16]. Mature DCs can process and present various antigenic substances, activating T lymphocytes through interactions with T cell receptors (TCRs) via their surface-expressed co-stimulatory molecules and MHC–antigen complexes [11,17]. Concurrently, they secrete tumor necrosis factor (TNF)-α, interleukin-(IL)-6, IL-23, IL-12, and other cytokines that accelerate the polarization of CD4^+^ T cells to Th1 and Th17 subsets with pro-inflammatory effects [18,19]. Research has demonstrated that natural small molecules derived from herbal sources can modulate the activation and function of T cells by inhibiting DC maturation, thereby offering potential therapeutic avenues for autoimmune diseases, including RA.

In recent years, natural small-molecule compounds have been recognized as important sources for the development of new drugs due to their multiple biological activities and minimal side effects. Studies have reported that small-molecule drugs derived from plants, such as resveratrol [20], epigallocatechin gallate (EGCG) [21], cyanidin-3-O-glucoside (C3G) [22], and cannabinoids [23], can alleviate inflammation and modulate immune inflammation responses and have been utilized in the treatment of incurable diseases, such as RA and SLE. PIC, a hydroxylated derivative of resveratrol [24], also possesses various biological activities. Gao X et al. [25] found that PIC can restrain the inflammatory response of RA-FLS and promote cell apoptosis by inhibiting the activation of classical inflammatory signaling pathways, such as the NF-κB and MAPK pathways, thereby improving collagen-induced arthritis in rats. However, investigations into the other biological properties of PIC have not been documented, and research on whether it can treat rheumatoid arthritis by exerting an immunosuppressive effect to regulate the immune balance remains unexplored.

To fill this gap, we identified the immunosuppressive effect of PIC through the DC cell model induced by LPS in vivo and in vitro. We constructed the AIA mouse model using CFA and IFA to explore the immunomodulatory function of PIC in vivo. Network pharmacology, molecular docking, and Western blotting were employed to clarify the mechanisms by which PIC modulates the immune balance. In general, our research aimed to clearly probe the function and mechanisms of PIC targeting DCs in the treatment of rheumatoid arthritis.

## 2. Results

### 2.1. PIC Inhibits LPS-Induced DC Maturation and Function In Vitro

Given the central role of DCs in initiating autoinflammatory immune responses in arthritis, we conducted experiments to verify the function of PIC on DCs in vitro. First, we assessed the toxicity range of the small-molecule compounds derived from the herb by exploring the viability of the DCs. Our results indicated that the concentration of PIC below 2 μM did not significantly affect the viability of the DCs (*p* > 0.05) (Appendix A). Second, when the DCs were treated with PIC alone within this concentration range, there was no significant difference in the expression of co-stimulatory molecules, or in the secretion levels of cytokines, compared to the untreated group (*p* > 0.05) (Appendix A). However, at this concentration, PIC significantly reduced the expression of CD40, CD86, and MHCII (*p* < 0.01) (Figure 1A) and decreased the levels of IL-12p40 and IL-6 (*p* < 0.01) (Figure 1B) of the DCs in response to LPS stimulation, while simultaneously increasing the secretion of IL-10 and TGF-β (*p* < 0.001) (Figure 1C). The optimal inhibitory concentration of PIC was determined to be 2 μM.

Following stimulation by inflammatory mediators, DCs gradually mature and lose their ability to internalize antigens. Therefore, we next examined the phagocytosis of antigens by DCs in the presence of PIC. Using dextran as a model antigen, which can be visually observed when co-cultured with fluorescently labeled dextran, the PIC + LPS treatment group receiving PIC and LPS exhibited a significant increase in the double expression of CD11c^+^Dextran^+^ compared to the LPS-only group. This suggested that PIC could enhance the antigen uptake capability of DCs (*p* < 0.01) (Figure 1D) while concurrently inhibiting their maturation process. Furthermore, since DCs also initiate their migration to draining lymph nodes following maturation through the high expression of CCR7 [26], we examined the impact of PIC on the expression of CCR7 on the surfaces of DCs. The results demonstrated that PIC exhibited a remarkable reduction in the CCR7 expression compared to the LPS group (*p* < 0.001) (Figure 1E).

Additionally, results from laser confocal microscopy showed that the DCs in the LPS group exhibited a pronounced maturation phenotype, characterized by the presence of dendritic protrusions. In contrast, the DCs in both the normal and drug treatment groups displayed a rounded or oval morphology (Figure 1F). Collectively, these results indicate that PIC can inhibit the LPS-induced maturation of DCs and influence their morphology.

### 2.2. PIC Inhibited LPS-Induced DC Maturation and Function In Vivo

To investigate whether the effects of PIC on the DCs observed in vitro also occur in vivo, we injected various concentrations of PIC mixed with LPSs into the paws of mice and performed flow cytometry analysis on the lymph nodes of the right legs 24 h post-injection. The results demonstrated that the PIC treatment group had markedly reduced proportions of CD11c^+^CD40^+^, CD11c^+^CD86^+^, and CD11c^+^MHCII^+^ cells (*p* < 0.01) (Figure 2A). Concurrently, DCs treated with the drug in vitro were CFSE-stained and subsequently injected into the feet of mice, from which were isolated lymphocytes from the draining lymph nodes for analysis. We observed that the proportion of cells expressing CD11c^+^CFSE^+^ in the PIC + LPS group was markedly lower than that in the LPS group (*p* < 0.05) (Figure 2B). In summary, PIC can suppress the inflammation-induced maturation and migration of DCs, thereby regulating the activation of cellular immune responses in vivo.

### 2.3. Evaluation of PIC’s Therapeutic Effect on the AIA Mouse Model

The inhibitory effect of PIC on the maturation and migration of DCs prompted us to check the mitigating effect of this traditional drug on autoimmune diseases. To this end, we established an autoimmune arthritis mouse model by injecting CFA. Following the construction of the arthritic model, as outlined in the flowchart in Figure 3A, the mice were treated with varying doses of PIC, with methotrexate (MTX) serving as the positive control for inflammatory arthritis. After 11 days of treatment, we observed that PIC did not have a significant effect on the overall body weights and organ indexes of the mice (*p* > 0.05) (Appendix A). Furthermore, no toxic effects were detected in the mice based on liver and kidney function tests (*p* > 0.05) (Appendix A). In comparison to the untreated group, the degree of swelling in the paws and ankle joints of the mice in the arthritic model group was conspicuously increased (*p* < 0.001), confirming the successful establishment of the arthritic model using CFA. Notably, the high-dose PIC treatment group exhibited significant reductions in their paw widths, ankle joint widths, and thicknesses compared to the model mice (*p* < 0.001) (Figure 3B,C). A histopathological examination of the joint tissues showed that mice in the normal control group exhibited intact and smooth joint structures, with no evidence of inflammatory cell infiltration. In contrast, mice in the model group displayed cartilage surface fibrosis and synovial hyperplasia, which invaded the joint cavity, resulting in joint space narrowing and accompanied by inflammatory cell infiltration. However, mice in the high-dose treatment group demonstrated widened joint spaces, reduced inflammatory cell infiltration, decreased synovial hyperplasia, and smoother surfaces, characterized by orderly arrangements of chondrocytes and osteocytes (Figure 3D). Furthermore, an ELISA of the cytokines in the mouse sera indicated high levels of (*p* < 0.001) IL-17A, IFN-γ, TNF-α, and IL-6 in the model group. Interestingly, PIC treatment led to a reduction in these inflammatory cytokines and an obvious increase in the IL-10 levels (*p* < 0.001) (Figure 3E).

To further reveal the mechanism underlying the relieving effect of PIC on autoinflammatory arthritis, flow cytometry was employed to examine the activation of relevant immune cells in the lymph nodes and spleens of mice. Following PIC treatment, we observed that the number of mature DCs expressing CD40 (CD11c^+^CD40^+^) and CD86 (CD11c^+^CD86^+^) significantly decreased (*p* < 0.001) (Figure 4A). In addition, high-dose PIC significantly reduced the levels of TNF-α, IL-23, and IL-6 (*p* < 0.001) secreted by DCs in the spleens (Figure 4B). Moreover, compared to the model group, high-dose PIC significantly restrained the polarization of CD4^+^ T cells into Th1 and Th17 cells. It also notably increased the proportion of Th2 cells and promoted the production of inducible regulatory T cells (iTreg) and natural regulatory T cells (nTreg) (*p* < 0.001) (Appendix A). The ratios of Th2/Th1, nTreg/Th17, and iTreg/Th17 were significantly elevated (*p* < 0.001) (Figure 4C). Interestingly, in the mouse arthritis model, we found that IL-23 and IL-6 or TNF-α secreted by DCs were positively correlated with Th17 or Th1 cells (*p* < 0.001) (Figure 4D), indicating that PIC may inhibit the polarization of CD4^+^ T cells into pro-inflammatory Th1 and Th17 cells by impeding DCs’ production of inflammatory cytokines. Based on these findings, we concluded that PIC can ameliorate foot and joint damage in mice by inhibiting the activation of DCs and regulating the polarization of CD4^+^ T cells in the spleens of CFA-induced arthritic mice, thereby alleviating inflammation and restoring the immune balance.

### 2.4. PIC-Treated DCs Inhibited Th17 and Upregulated Treg Differentiation from T Cells in Experimental Arthritis Mouse Model

The effect of PIC on the DC-stimulated proliferation of OVA-specific T cells was explored in vitro. In the presence or absence of LPSs, DCs were treated with PIC and then co-cultured with spleen cells from OTII mice for 72 h. After staining with CFSE, it was observed that the DCs co-treated with PIC and LPSs had markedly reduced proportions of CD4^+^CFSE^−^ T cells compared to the LPS-only group (*p* < 0.001) (Figure 5A).

Furthermore, to confirm the effect of the drug-treated DCs on the T cell differentiation in the spleens of arthritic mice, DCs treated with PIC alone or in combination with LPSs were co-cultured with spleen cells from the mice of the CFA-induced arthritic model of the same strain. The PIC + LPS group significantly decreased the proportions of CD4^+^IFN-γ^+^ and CD4^+^IL-17A^+^ cells, while increasing the proportions of CD4^+^IL-4^+^, CD4^+^CD25^−^Foxp3^+^, and CD4^+^CD25^+^Foxp3^+^ cells (*p* < 0.01) (Figure 5B). These findings indicate that the PIC-treated DCs inhibited not only the proliferation of OVA-specific CD4^+^ T cells but also the polarization of CD4^+^ T cells in the CFA-induced mouse model.

### 2.5. Network Pharmacology and Molecular Docking

To probe the immunosuppressive mechanism of PIC in the treatment of RA, we employed network pharmacology prediction and molecular docking techniques. Initially, we screened 92 compound targets using the TCMSP database, the Swiss Target Prediction database, and the PharmMapper database. Subsequently, from the DrugBank database and Gene Card database, we retrieved 1334 targets associated with rheumatoid arthritis. By intersecting these targets, we identified a total of 47 cross-targets, which were designated as the key targets of PIC in combating RA (Figure 6A). We then conducted a protein–protein interaction (PPI) network analysis on these key targets, revealing an average degree value of 14.808 and an average clustering coefficient of 0.465. Utilizing the CytoNCA plug-in, we further filtered the targets to retain those exceeding the average degree value, resulting in a total of 19 key targets (Figure 6B,C). In the graphical representation, the sizes and colors of the nodes vary according to their degree values.

The functional enrichment analysis of the key targets revealed that the biological processes were predominantly enriched in the regulation of inflammatory and immune responses, the regulation of the MAPK cascade, the positive regulation of cell migration, the positive regulation of T cell activation, and the regulation of cell proliferation in the Gene Ontology (GO) enrichment analysis. The molecular functions were primarily associated with transcription factor binding, growth factor binding, ATPase binding, and nuclear receptor activity. The cellular components were mainly related to plasma membrane microdomains, vesicle cavities, receptor complexes, and cell–substrate connections (Figure 6D). Furthermore, 15 key targets were involved in the P13K-Akt, MAPK, NF-κB, IL-17, TNF, chemokine, and other related signaling pathways in the Kyoto Encyclopedia of Genes and Genomes (KEGG) enrichment analysis (Figure 6E,F). Cytochalasin D inhibits the phagocytosis of DCs mainly by interfering with microfilament polymerization and disrupting the dynamic structure of the cytoskeleton. Interestingly, we found that the addition of cytochalasin D (0.5 μg/mL) alongside simultaneous treatment with PIC and LPSs eliminated the inhibitory effect of PIC on the DCs, showing no significant difference when compared to the LPS group (*p* > 0.05) (Figure 7A). This suggested that PIC can be internalized by DCs to exert its immunosuppressive function.

To confirm that the analysis conducted through network pharmacology accurately reflected the conditions in DCs, we examined the relevant signaling pathways within DCs. The excessive activation of pathways such as MAPK and NF-κB can lead to the production of numerous pro-inflammatory mediators and cytokines, resulting in synovial inflammation and hyperplasia in the joints [27,28]. Similarly, we speculated that PIC may inhibit the maturation of DCs through the MAPK and NF-κB signaling pathways, thereby serving as a potential therapeutic agent for RA. According to the KEGG enrichment analysis, we identified MAPK14, PTGS2, KIT, EGFR, and RELA as the intracellular targets associated with these signaling pathways. We subsequently performed molecular docking simulation with the compound PIC, using the corresponding target inhibitors as positive controls. The results showed that the binding affinities of PIC to MAPK14, PTGS2, KIT, EGFR, and RELA was comparable to or exceeded those of the positive control drugs, with binding affinities of −8.9, −8.4, −8.8, −8.1, and−8.3 kcal/mol, respectively, indicating that PIC exhibited a strong interaction with the selected targets (Figure 7B). The docking visualization analysis showed that PIC engaged in various binding interactions with the target proteins, including Pi–Pi T-shaped, Pi–cation, Pi–alkyl, Pi–Pi-stacked, Pi–sigma, and hydrogen bonds. We further examined whether the drug treatment influenced the activation of the MAPK and NF-κB signaling pathways using Western blot analysis. The results showed that the treatment of DCs with PIC for 6 h inhibited the LPS-induced phosphorylation of ERK, p38, and JNK, and also inhibited the nuclear translocation of p65 in the NF-κB signaling pathway (*p* < 0.05) (Figure 7C). Therefore, these results verified that PIC can suppress the activation of the MAPK and NF-κB signaling pathways, thereby reducing inflammatory responses.

## 3. Discussion

The pathogenesis of rheumatoid arthritis (RA) is extremely intricate, with pathogenic T cells playing an immediate role in the onset and development of the disease [29]. According to research reports, the infiltration of CD4^+^ T cells at sites of inflammation is a defining characteristic of autoimmune diseases. In the early stages of RA, the synovial tissue of affected joints is populated by memory CD4^+^ T cells [30], which can further polarize into different phenotypes depending on the cytokine environment, thereby participating in the immune response [31]. In addition, DCs are among the first cells to be activated in the synovial tissue of RA patients.

DCs act as a crucial bridge between the innate and adaptive immune systems, playing a vital role in maintaining immune tolerance within the body [32]. Immature DCs serve as sentinels of the immune system, primarily recognizing antigens through pattern recognition receptors and possessing a high phagocytic capacity. However, when stimulated by inflammatory mediators resulting from self-tissue damage, necrotic cells, or pathogens, immature DCs undergo maturation, which is characterized by elevated levels of MHC and co-stimulatory molecules [33,34]. This study found that under LPS-induced inflammatory conditions, PIC can downregulate the expression of surface molecules on DCs and reduce their secretion of inflammatory cytokines, thereby significantly inhibiting the maturation of LPS-induced DCs. Mature DCs, which express high levels of the chemokine receptor CCR-7, migrate to lymph nodes by interacting with CCL19 and CCL21, where they function in areas enriched with T cells [35,36]. Our results indicated that PIC inhibited the migration of DCs from peripheral tissues to the lymph nodes, consequently suppressing the activation and differentiation of T lymphocytes. Moreover, mature DCs possess a robust ability to present antigens; they process self-reactive antigens into peptides and present them to T cells while secreting IL-12, TNF-α, and TGF-β, which contribute to the activation of CD4^+^ T cells and their further differentiation into pro-inflammatory effector cells involved in inflammatory responses and joint damage [37,38]. However, the activation and differentiation of CD4^+^ T cells were inhibited by DCs treated with PIC. Furthermore, the immunosuppressive effect of PIC was also identified in the AIA mice model, and similar effects were observed in vitro.

PIC, as a hydroxylated derivative of resveratrol, is derived from various plants [39]. Like resveratrol, PIC possesses potent antioxidant activity, as well as anti-inflammatory and anticancer properties [40]. This study demonstrated that PIC not only inhibited the maturation of DCs but also suppressed T cell activation. Additionally, PIC has been shown to ameliorate inflammation by inhibiting the TLR4/NF-κB/NLRP3 pathway in hepatic macrophages [41]. Furthermore, piceatannol is more effective than resveratrol at suppressing adipogenesis in human visceral adipose-derived stem cells due to its distinct antilipolytic properties [42,43]. PIC possesses the ability to reduce the toxic action of neutrophils through the involvement of protein kinase C [44].

Methotrexate (MTX) is currently the first-line treatment for RA; however, prolonged use may lead to a range of adverse effects [45,46]. Our results indicated that the serum levels of pro-inflammatory cytokines after PIC treatment were lower than those in the MTX group. In addition, PIC exhibited diverse physiological activities through its multi-pathway and multi-target properties. In this study, we used network pharmacology to predict that PIC could interact with targets related to DC maturation and function. PIC interacted with MAPK14 and RELA to regulate the activation of the NF-κB and MAPK signaling pathways in DCs by molecular docking and Western blotting. PIC can enhance therapeutic efficacy by inhibiting DC maturation via multiple targets. However, given that the MAPK pathway is implicated in numerous cellular life processes, the inhibition of this pathway by PIC may lead to potential off-target effects. Therefore, further investigations are warranted to comprehensively assess the safety profile of PIC.

Of course, our study has several limitations that require further investigation. In the AIA mice model, the duration was short, and the changes in the immune homeostasis in the mice after long-term treatment were not observed. Moreover, a systematic assessment of PIC safety is lacking, which will be further investigated in future studies.

## 4. Materials and Methods

### 4.1. Materials

Female BALB/c mice aged 6–8 weeks were purchased from the Experimental Animal Center of Xinjiang Medical University. The animal experiment was conducted under the guidance of the Animal Care and Use Committee of the College of Life Science and Technology, Xinjiang University (XJUAE-2024-041).

Main Reagents: PIC (≥98%), Cytochalasin D (Yuanye, China); penicillin–streptomycin solution, complete Freund’s adjuvant (CFA), RPMI-1640 medium, phosphate-buffered saline (PBS) (Gibco, Carlsbad, CA, USA); fetal bovine serum (FBS) (Sijiqing, Hangzhou, China); granulocyte–macrophage colony-stimulating factor (GM-CSF) (Peprotech, NJ, USA); Lipopolysaccharides (LPSs); fluorescein isothiocyanate–dextran (FITC-Dextran); dimethyl sulfoxide (DMSO) (Sigma, St. Louis, MO, USA); mouse IL-12p40, IL-6, and IL-10 ELISA kits (Elabscience, Wuhan, China); fluorescence-labeled antibodies against CD4-FITC, CD11b-FITC, CD11c-APC, CD86-PE, F4/80-PB450, CD40-APC, CD3-APC, etc. (BD Biosciences, Franklin Lakes, NJ, USA); GATA-PerCP/Cyanine, FOXP3-PerCP/Cyanine 5.5, RORγt-PB450, T-Bet-PerCP/Cyanine 7, IL-23-APC, TNF-α-ER780, etc. (BD Biosciences, Franklin Lakes, NJ, USA); Western blot antibodies such as anti-p38, JNK, ERK, etc. (Cell Signaling Technology, Danvers, MA, USA).

Main Instruments: high-speed refrigerated centrifuge (Thermo Fisher Scientific, Waltham, MA, USA); microplate reader (Bio-Rad, Hercules, CA, USA); CytoFlex (Beckman Coulter, Pasadena, CA, USA); CO_2_ cell culture incubator (Thermo Fisher Scientific, Waltham, MA, USA); Confocal Laser Scanning Microscope (Nikon, Tokyo, Japan), etc.

### 4.2. Induction and Culture of Mouse Bone Marrow-Derived DCs

The methods reported in the literature [47] were used to induce and culture bone marrow-derived dendritic cells. Bone marrow cells were rinsed from the leg bones of mice using RPMI-1640 base medium (10% FBS + 1% PS). After filtration and centrifugation, single-cell suspensions were prepared and seeded into 60 mm cell culture dishes with RPMI-1640 medium (containing GM-CSF). The cells were incubated in the cell culture incubator (37 °C and 5% CO_2_) for 7 days, with fresh culture medium replaced on days 2, 3, and 5.

### 4.3. In Vitro Maturation of DCs

In the in vitro experiments, the method reported in the literature [48] was used to verify the inhibitory effect of PIC on the DC maturation. Cells collected on the 7th day of induced culturing were inoculated into 24-well cell culture plates (1 million per well) and treated with different doses of PIC (1, 1.5, and 2 μM) and LPSs (80 ng/mL) either in combination or individually for 12 h. After the collection and centrifugation of DCs, the supernatant was collected. The cell pellet was stained in the dark for 20 min with anti-CD40-APC and anti-CD86-PE antibodies, and the expression levels of the co-stimulatory molecules on the DC surfaces were detected using CytoFlex. Subsequently, the secretion of cytokines from DCs in the supernatant were determined by ELISA, following the specific instructions provided in the corresponding ELISA assay kit. Additionally, the expression of the cytokines in the DCs were also detected using flow cytometry. After 1 h of treatment according to the above method [47], Golgistop was added and incubated. After 12 h, the cells were collected by centrifugation, stained with anti-TGF-β-APC and anti-IL-10-PE antibodies, and subsequently analyzed using CytoFlex.

The Annexin V-FITC/PI cell apoptosis kit [49] was used to verify the effect of PIC at effective concentrations on the viability of the DCs.

To assess the antigen uptake capacity of the DCs, cells induced on the 7th day were treated with drugs and LPSs for 12 h, followed by the addition of FITC-Dextran and incubation for 1 h [50]. Subsequently, the reaction was terminated by adding pre-chilled PBS, after which the cells were collected and analyzed using CytoFlex.

In addition, the collected cells were seeded in a confocal dish and incubated for 12 h, washed with PBS and fixed with 4% paraformaldehyde, treated with 0.1% Trion X-100 at room temperature, followed by the addition of DAPI dye and FITC-phalloidin (Sigma, St. Louis, MO, USA) for 10 min to avoid light staining, and then the dish was observed under laser confocal scanning microscopy (Nikon, Tokyo, Japan).

### 4.4. In Vivo Maturation of DCs

Different concentrations of PIC (20, 40, and 80 mg/kg) were mixed with LPSs (100 ng/mouse) and injected into the paws of mice (25 μL/mouse). After 1 day, lymph nodes from the right leg popliteal area were harvested, followed by grinding and centrifugation to separate lymphocytes. The lymphocytes were then stained with CD11c-FITC, CD86-PE, CD40-PerCP, and MHCII-PC7. The samples were then analyzed using CytoFlex.

### 4.5. Validation of DC Migration Ability

The DCs were collected on the 7th day and treated with drugs and LPSs for 12 h. Then, PE-CCR7 antibodies were added for staining in the dark and the samples were analyzed using CytoFlex.

Additionally, after the DCs were treated with drugs and LPSs for 12 h, the cells were collected and resuspended at 1 × 10^7^ cells/mL, and 1 μM CFSE was added for dark staining for 10 min. This suspension was injected into the paws of the mice (50 μL/mouse). After 24 h, the inguinal lymph nodes of the legs were collected. After grinding and centrifugation, the lymphocytes were isolated and further stained with CD11c-FITC in the dark for 20 min. Finally, the samples were analyzed using CytoFlex.

### 4.6. PIC Inhibits the Proliferation of Specific CD4^+^T Cells

The proliferation of CD4^+^ T cells was detected by the method in the literature in vitro [51]. DCs cultured for 7 days were collected and divided into the untreated group, LPS treatment group (80 ng/mL), PIC (2 μM) + LPS treatment group, and PIC (2 μM) treatment group. After 12 h of drug and LPS treatment, the cells were collected by centrifugation for later use. At the same time, the spleens of the OTII mice were prepared as single-cell suspensions and treated with RBC Lysate for 3 min. After PBS washing twice, the medium was added to re-suspend, 5 mL of CFSE was added to each 1 × 10^7^ cell for 20 min, and then the pre-cooled PBS was added to terminate the reaction, and the cells were collected by centrifugation. Subsequently, the above DCs were mixed with spleen cells at a ratio of 1:5 and 0.1 mg/mL OVA antigen was added. The cells were collected after 72 h, stained with CD3-APC-A700 and Percp-CD4-PerCP/Cyanine 5.5 flow antibodies in the dark, and detected by flow cytometry.

### 4.7. DCs Treated with PIC Inhibited T Cell Differentiation

After the emulsification of PBS and CFA at the ratio of 1:1, they were injected into the paws of mice (25 μL/mouse). After 48 h, DCs treated by PIC and LPS for 12 h were co-cultured with spleen lymphocytes at a ratio of 1:5. They were further cultivated for 72 h after Golgistop was added, after 72 h, the cells were collected and stained with IL-17-PE, IL-4-PerCP/Cyanine 5.5, CD3-APC, and CD4-FITC. Finally, the samples were analyzed using a CytoFlex flow cytometer.

### 4.8. Network Pharmacology and Molecular Docking

Compound targets were predicted through the TCMSP database (URL: https://tcmsp-e.com/, accessed on 15 November 2023), Swiss Target Prediction database (URL: http://www.swisstargetprediction.ch, accessed on 16 November 2023), and PharmMapper database (URL: https://www.lilab-ecust.cn/pharmmapper/, accessed on 16 November 2023). Disease targets related to RA were retrieved through the DrugBank database (URL: https://go.drugbank.com/, accessed on 16 November 2023) and Gene Card database (URL: https://www.genecards.org/, accessed on 16 November 2023). VENNY 2.1 (URL: https://bioinfogp.cnb.csic.es/tools/venny/, accessed on 17 November 2023) was used to obtain the intersection of targets between PIC and RA disease. The key targets were imported into the String database (URL: https://string-db.org/, version 11.0, accessed on 17 November 2023) to obtain the protein interaction relevance, and the PPI network and “target-pathway” interaction network were constructed in Cytoscape 3.6.1 software based on degree values. KEGG and GO enrichment analyses were performed through Metascape (URL: https://metascape.org/gp/index.html#/main/step1, accessed on 18 November 2023) and the Microbio Online platform (URL: https://www.bioinformatics.com.cn/, accessed on 18 November 2023). The 3D structures of the compounds were retrieved from the PubChem database (URL: https://pubchem.ncbi.nlm.nih.gov/, accessed on 20 November 2023), and the PDB target protein crystal structure was downloaded from the UniProt database (URL: https://www.uniprot.org/id-mapping, accessed on 20 November 2023) and RCSB PDB (URL: https://www.rcsb.org/, accessed on 20 November 2023). The protein structures were processed using Chem3D and PyMOLWin software (1.0.0.0) and were then imported into AutoDock Vina software (1.5.7) for molecular docking simulation with compounds. Docking visualization analysis was performed using Discovery Studio software (19.1.0.18287).

### 4.9. Western Blots

DCs induced on the 7th day were collected and stimulated with or without LPSs and SOG for 6 h. Proteins were extracted and quantified using a BCA kit. Subsequently, SDS-PAGE was performed at a constant voltage, and the proteins were transferred to the PVDF membrane. The membrane was placed in 5% skimmed milk/BSA solution and incubated for 2 h. The membrane was incubated in the diluted primary antibodies after washing with PBST. After the unbound primary antibodies were washed off with PBST, the membrane was incubated with the corresponding conjugated secondary antibodies. After washing with PBST again, the ECL chemiluminescence substrate kit was used for imaging.

### 4.10. CFA-Induced Mouse Model of Arthritis

According to the method outlined in Reference [52], 30 mice were randomly divided into the untreated, model, methotrexate, and PIC (10, 20, 40 mg/kg) treatment groups. After the emulsification of PBS and CFA at the ratio of 1:1, they were injected into the paws of mice (25 μL/mouse), followed by a booster injection administered after 14 days. After 2 weeks, the mice were treated with PIC or methotrexate every other day, and the treatment was ended on the 11th day. During the treatment period, the joint index and body weight were recorded every other day. Prior to euthanasia, serum was collected through retro-orbital bleeding, and the levels of relevant inflammatory cytokines were determined according to the ELISA kit instructions.

### 4.11. Data Analysis

All flow cytometry data were analyzed using CytExpert software (2.4.0.28). Other data were analyzed using GraphPad Prism 9.0 statistical software. Statistical significance was determined using one-way analysis of variance (ANOVA) or unpaired *t*-test, with *p*-values *<* 0.05 considered statistically significant.

## 5. Conclusions

In general, we demonstrated that PIC exhibited an anti-inflammatory function in the AIA mice model by suppressing the maturation and secretion of pro-inflammatory cytokines from DCs in the lymph nodes and spleens, further restraining the differentiation of Th17 and Th1 cells in the spleens, so as to decrease the levels of IL-6, TNF-α, and IL-17A, and so on in peripheral blood. The combined analysis of network pharmacology and in vitro experiments further proved the immunosuppressive effect of PIC via the suppression of the maturation and immune activation function of DCs via the MAPK and NF-κB signaling pathways, which was further validated by molecular docking and Western blotting. Our research proved the immunoregulating function of PIC on the maturation and function of DCs in the AIA mice model; thus, PIC has potential application in the treatment of RA patients.

## Figures and Tables

**Figure 1 ijms-26-03626-f001:**
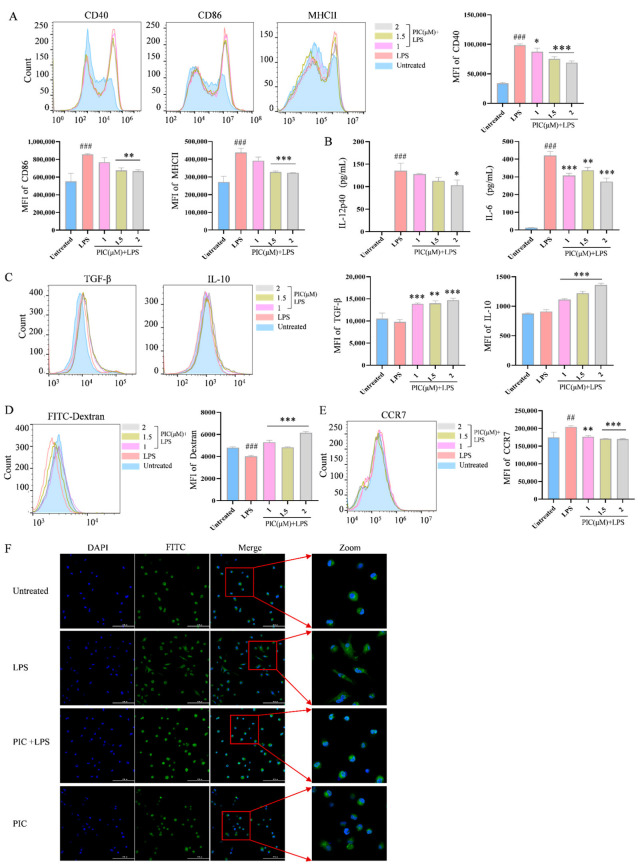
The effects of PIC on LPS-induced DC maturation and function in vitro. DCs were treated with varying doses of PIC in the presence or absence of LPSs and detected by flow cytometry and ELISA: (**A**) the expression levels of CD40, CD86, and MHC II on the surfaces of DCs; (**B**) the secretion levels of cytokines IL-12p40 and IL-6; (**C**) cytokine TGF-β and IL-10 expression; (**D**) dextran co-cultured with DCs to detect the phagocytosis of antigens; (**E**) the expression of CCR7 on DC surface; (**F**) the morphology of DCs under the laser confocal microscope (Scale bar is 100 μm). Statistical significance was noted as ^##^ *p* < 0.01 and ^###^
*p* < 0.001 compared to the untreated group, and as * *p* < 0.05, ** *p* < 0.01 and *** *p* < 0.001 compared to the LPS group.

**Figure 2 ijms-26-03626-f002:**
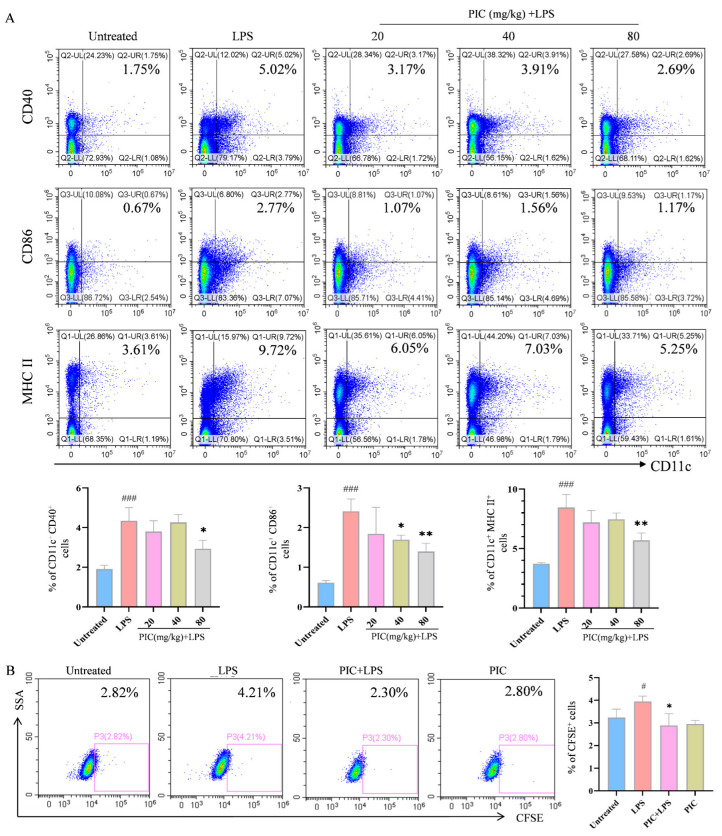
The effects of PIC on the maturation and migration of DCs in vivo. PIC (20, 40, and 80 mg/kg) was mixed with or without LPS and injected into the paws of mice for 24 h: (**A**) the proportions of CD11c^+^CD40^+^, CD11c^+^CD86^+^, and CD11c^+^MHCII^+^ cells in the lymph nodes of the right legs; (**B**) the proportions of DCs stained with FITC-CFSE in lymph nodes. ^#^ *p* < 0.05, ^###^ *p* < 0.001 compared to the untreated group. * *p* < 0.05 and ** *p* < 0.0 compared to the LPS group.

**Figure 3 ijms-26-03626-f003:**
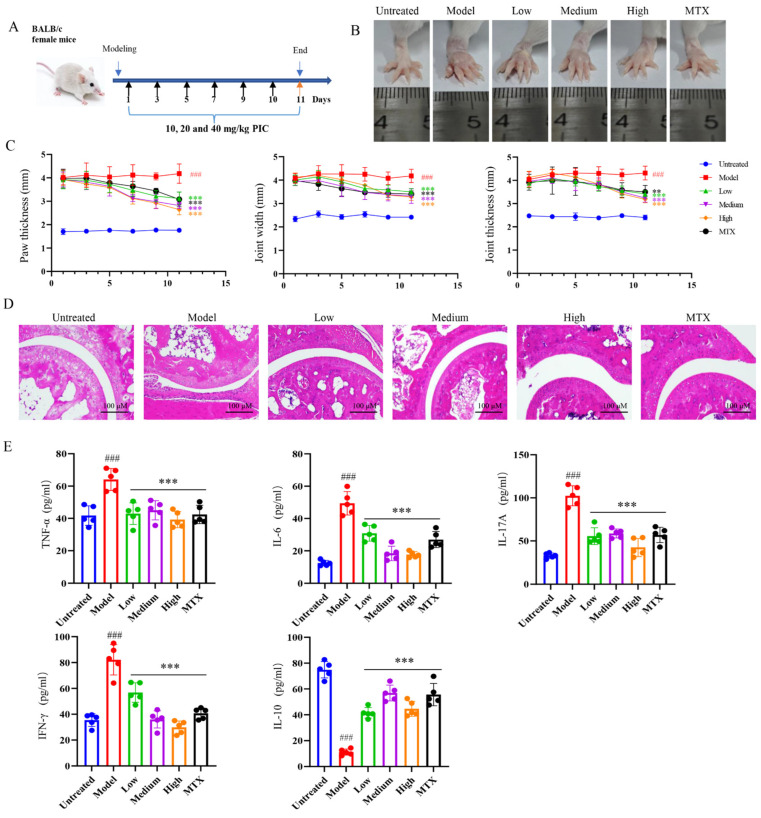
Therapeutic effects of PIC on arthritis in mice. After the CFA-induced arthritic mouse model was successfully constructed, PIC or MTX was injected subcutaneously every other day. (**A**) Flowchart of arthritic mouse modeling and treatment; (**B**) swelling degree of mouse foot and ankle joints; (**C**) from the first day of treatment, the foot and ankle joints of mice were measured every other day; (**D**) joint pathology histological section stained with H&E; (**E**) the expression levels of cytokines TNF-α, IL-6, IL-17 A, IFN-γ, and IL-10 in sera of mice were detected by ELISA. ^###^ *p* < 0.001 compared to the untreated group. ** *p* < 0.01 and *** *p* < 0.001 compared to the LPS group.

**Figure 4 ijms-26-03626-f004:**
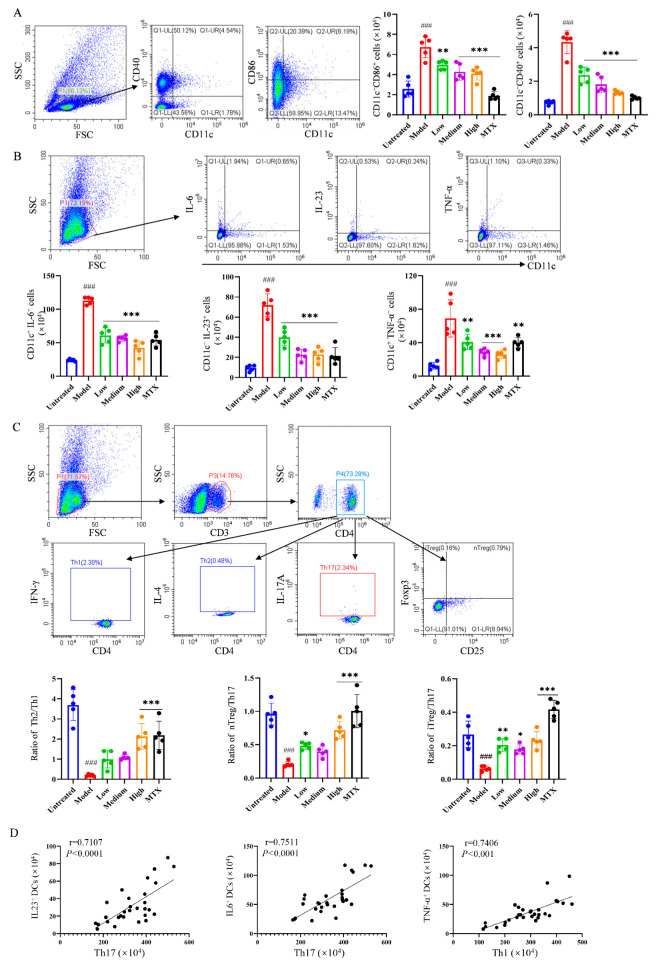
The effects of PIC on immune cells in lymph nodes and spleens of arthritic mouse model. (**A**) The expression levels of DC co-stimulatory molecules CD40 and CD86 in mouse leg lymph nodes were detected; (**B**) the levels of the cytokines TNF, IL-23, and IL-6 secreted by DCs in the spleens of mice were detected; (**C**) different cytokines secreted by Th1, Th2, and Th17 in the spleens of mice and the percentages of Th2/Th1, nTreg/Th17, and iTreg/Th17 were analyzed; (**D**) the correlations between IL-23-secreting- and IL-6-secreting DCs with Th17 cells and TNF-α-secreting DCs with Th1 cells in arthritic mice in vivo. ^###^ *p* < 0.001 compared to the untreated group. * *p* < 0.05 ** *p* < 0.01 and *** *p* < 0.001 compared to the LPS group.

**Figure 5 ijms-26-03626-f005:**
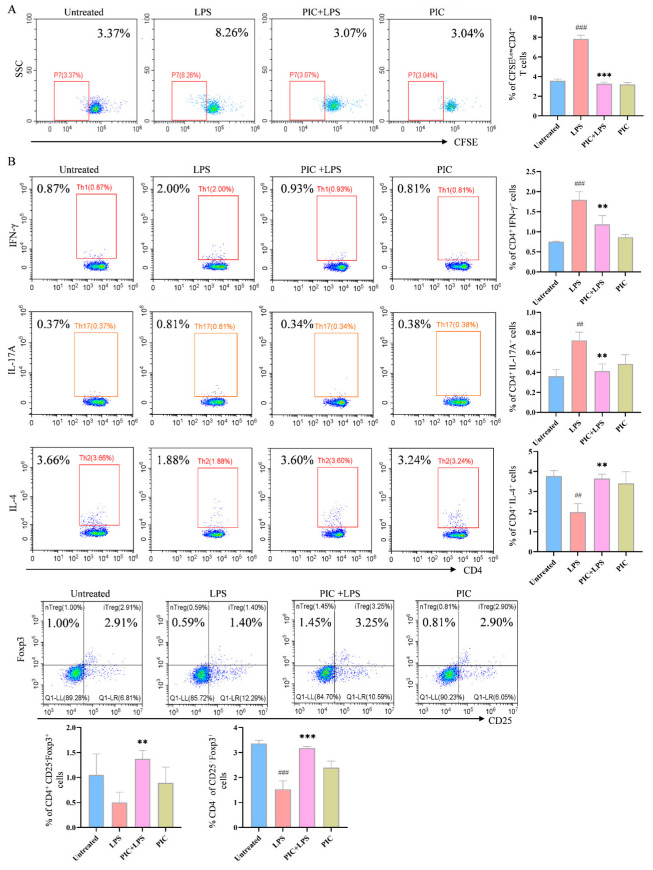
PIC inhibited the DC-stimulated proliferation of OVA-specific CD4^+^ T cells and the differentiation of T cells from the spleens of arthritic mice in vitro. (**A**) DCs were treated with PIC in the presence of absence of LPSs for 12 h, mixed with CFSE-labeled OT II mouse spleen cells at a ratio of 1:5 before 0.1 mg/mL OVA antigen was added, and co-cultured in 24-well plates. The effect of PIC on the proliferation of specific CD4^+^ T cells was detected after 72 h. (**B**) The treated DCs were mixed with spleen cells from arthritic mice at a ratio of 1:5. After 72 h of co-culturing in 24-well plates, the effect of the PIC on the polarization of CD4^+^ T cells into the Th1, Th2, Th17, and Treg cell subtypes was detected by flow cytometry. ^##^ *p* < 0.01, ^###^ *p* < 0.001 compared to the untreated group. ** *p* < 0.01 and *** *p* < 0.001 compared to the LPS group.

**Figure 6 ijms-26-03626-f006:**
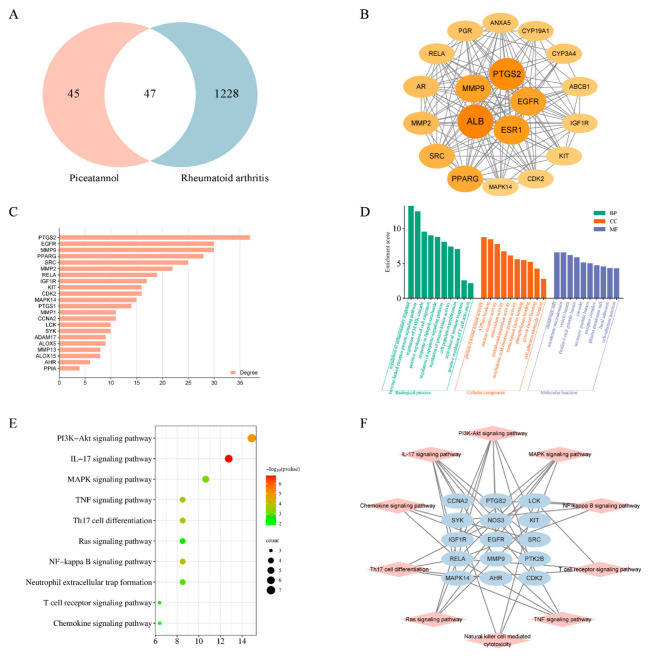
Network pharmacology prediction and molecular docking technology. (**A**) Venn diagram showing the intersection target between the active ingredient and the disease; (**B**,**C**) the protein interaction network diagram greater than the average degree value in the common targets of PIC and RA; (**D**) GO functional enrichment analysis; (**E**) analysis of KEGG pathway enrichment; (**F**) a network showcasing the primary pathways and targets involved in the treatment of RA with PIC.

**Figure 7 ijms-26-03626-f007:**
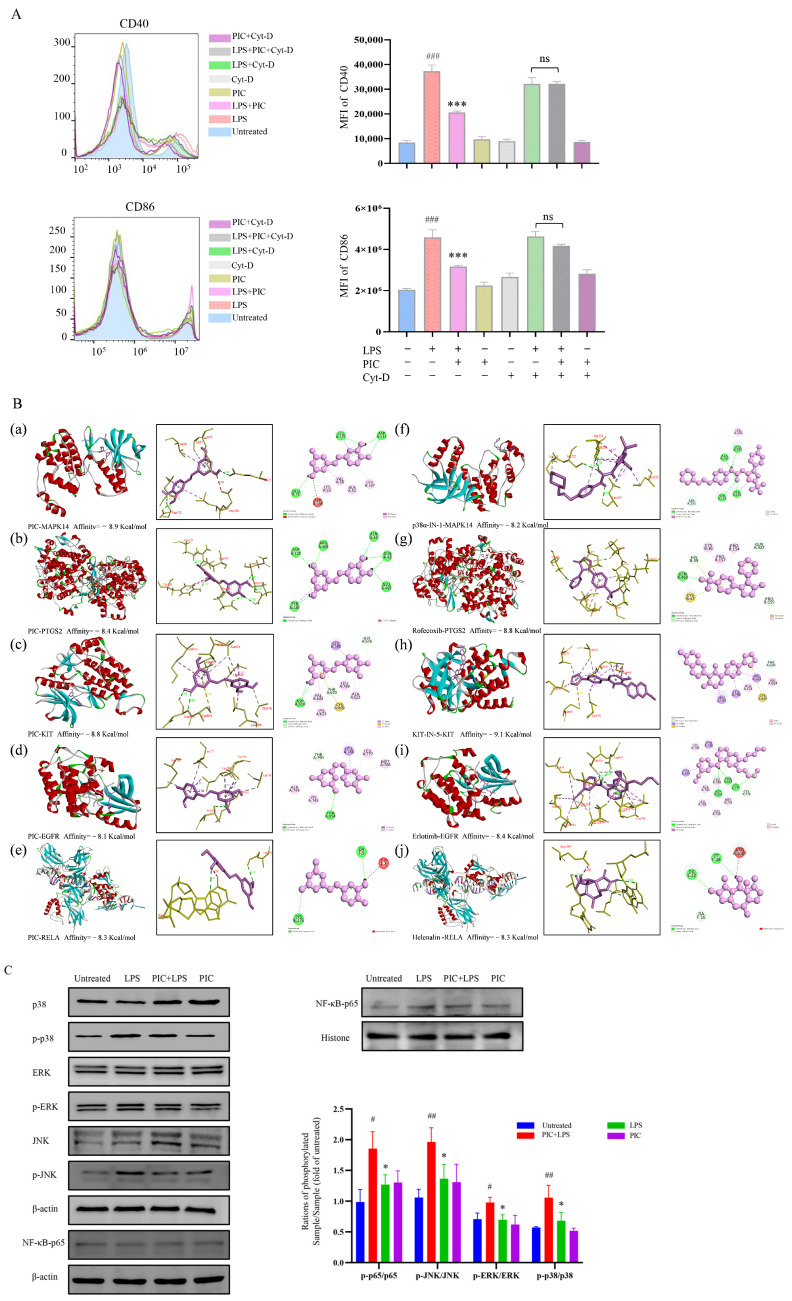
(**A**) Cytochalasin D inhibits the phagocytosis of DCs by interfering with the polymerization of microfilaments and destroying the dynamic structure of the cytoskeleton. We verified the endocytosis of PIC by DCs with or without cytochalasin D; (**B**) molecular docking of drugs and targets: (**a**–**e**) molecular docking of PIC with MAPK14, PTGS2, KIT, EGFR, and RELA; (**f**–**j**) molecular docking between PIC and the corresponding inhibitors for each target; (**C**) effects of drugs on the MAPK and NF-κB signaling pathways in DCs. Statistical significance is indicated as follows: ^#^ *p* < 0.05, ^##^ *p* < 0.01, ^###^ *p* < 0.001 compared to the untreated group; * *p* < 0.05 and *** *p* < 0.001 compared to the LPS group, ns compared to LPS and Cy-D group.

## Data Availability

The datasets used and analyzed during the current study are available from the corresponding author upon reasonable request.

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
