# Peer review of "Piceatannol Inhibits the Immunostimulatory Functions of Dendritic Cells and Alleviates Experimental Arthritis"

_ijms, 2025, doi:10.3390/ijms26083626_

Round 1
Reviewer 1 Report
Comments and Suggestions for Authors
This manuscript presents an interesting study evaluating the effect of piceatannol, a naturally active ingredient derived from Iris halophila Pall., on the immune response in a murine model of experimental arthritis. Specifically, the study observed that piceatannol restrained the maturation and function of dendritic cells and suppressed the activation and polarization of CD4+ T cells by inhibiting MAP kinases and NF-κB signalling, thereby alleviating experimental arthritis. The research topic is of considerable importance for human immunology and disease management, and the results appear promising for the treatment of rheumatoid arthritis, with an acceptable experimental design, data analysis, and well-presented figures. However, several significant deficiencies compromise the overall academic quality and value of the manuscript. Notably, the Discussion section is a complete repetition of the Introduction and Results, lacking any substantive discussion. The logic within the Introduction is also confusing and requires restructuring. Numerous formatting issues are also present. Specific comments are detailed below.
L16-19 - Animal design should be provided in the section.
L18 - Please define abbreviations when they first appear, DCs, CFA, MAP...
L19-24 - Results are too general, it is better to present the trend of changes and accurate values.
L27 - Formatting issue, regardless of being italic or not, “in vivo” is a phrase and the format should be consistent. This appears throughout the manuscript, please check and revise.
L28 - Keywords can expand more, such as inflammation, immunomodulation, etc.
L36-37 - Please provide at least 1-2 specific examples for each type of these drugs, general categories are not enough.
L37 - Abbreviations, NSAIDs, DMARDs. Please check throughout the paper.
L45 - Punctuation error, “T、B lymphocytes”.
L77-78 - This currently reads more like an abstract or conclusion. As an introduction, it should provide the background, purpose, potential outcomes, and broader implications of the research.
General comment for Introduction - The structure of this part is somewhat chaotic: problems, existing solutions and shortcomings, problem targets, and then new solutions. The suggestion is to immediately follow the problem introduction with potential targets, and then highlight the shortcomings of existing drugs and introduce the advantages and characteristics of this study.
L92-95 - This is the results section not a discussion or conclusion, a brief summary is not enough.
Please provide the specific numerical values of the indicator changes, as well as the p-value. Please check and revise this issue throughout the manuscript.
L204 - Formatting issue, “we explored...”
L290-331 - These contents overlap with the Introduction section. It is accepted to briefly mention the background and purpose of the study, but here is too long, too boring.
L332 - “The current study”, for yours or others? I suppose that here should be “this study” or “the present study”.
L332-361 - These contents overlap with the Result section, all of which are brief conclusions without any discussion. The discussion section should showcase how the authors derived these conclusions from data and image results.
L365-367 - Please provide the approval code and validity period.
L518 - Abbreviations used in the manuscript are not fully provided.
Author Response
Subject: Letter of major revision
Manuscript ID: ijms-3473583
Title: Piceatannol inhibits the immunostimulatory functions of dendritic cells and alleviates experimental arthritis
Dear Reviewers:
We thank you very much for your patient review and important suggestions for our article. Those comments are all valuable and very helpful for revising and improving our paper, as well as the important guiding significance to our researches. We are very sorry to update the revised manuscript so late, because the adding of some necessary information. We have improved the parts that you mentioned.
In this revised version, we have addressed the concerns of the reviewers. An item-by-item response to the reviewer’s comments is enclosed, and the revision was marked in red fronts in the manuscript. We hope that these revisions successfully address concerns and requirements and that this manuscript will be accepted.
The main corrections in the paper and the responds to the reviewers’ comments are as following:
Reviewer #1
- Comment: L16-19-Animal design should be provided in the section.
Response: Dear reviewer, thank you very much for your suggestions. We have checked the L16-L19 and not found the describe of animal experiment, which is the scheme about the experiments in vitro and mechanisms of PIC. We further carefully reviewed the abstract and replenished the animal design at L15. The detailed design of animal experiments was described at the section of 5.10. The revised content is as follows:
“……
Abstract: Rheumatoid arthritis (RA) is a highly prevalent systemic autoimmune dis-ease. Recently, natural small molecules have been explored as alternative therapeutic agents. Iris halophila Pall as the traditional herbal medicine, it is rich in active ingredients with anti-inflammatory and immunomodulatory effects. In our previous study, LC-MS analysis revealed that piceatannol (PIC) is one of the primary active ingredients in the root of Iris tectorum. The purpose of this study was to explore the immunomodulatory effects of PIC on the maturation and function of dendritic cells as well as on experimental arthritis induced by complete freund’s adjuvant (CFA) and incomplete freund’s adjuvant (IFA).
……”
- Comment: L18 - Please define abbreviations when they first appear, DCs, CFA, MAP...
Response: Dear reviewer, thank you for pointing out our problems. We have thoroughly reviewed and defined the abbreviations.
- Comments: L19-24 - Results are too general, it is better to present the trend of changes and accurate values.
Response: Dear reviewer, thank you for your valuable feedback on our manuscript. We have replenished the trend of changes and accurate values about the results as follows:
“……
The immunosuppressive mechanism of PIC on dendritic cells (DCs) was investigated through a joint analysis of network pharmacology and Western blotting. Our findings revealed that under Lipopolysaccharide (LPS)-induced inflammatory conditions, PIC could restrain the maturation and function of DCs (P < 0.001) and decrease the secretion of inflammatory cytokines (P < 0.001) compared to the LPS group. Furthermore, PIC suppressed the activation and polarization of CD4+ T cells, resulting in a decreased proportion of Th1 and Th17 cells (P < 0.001), ultimately improving the symptoms of CFA-induced arthritis in comparison to the model group.
……”
- Comments: L27 - Formatting issue, regardless of being italic or not, “in vivo”is a phrase and the format should be consistent. This appears throughout the manuscript, please check and revise.
Response: Dear reviewer, thank you for pointing out our problems. we regret the error that occurred in the format of “in vivo”. We have conducted a comprehensive examination and adjustment of the manuscript.
- Comments: L28-Keywords can expand more, such as inflammation, immunomodulation, etc.
Response: Dear reviewer, thank you for your valuable feedback on our manuscript. We have
replenished the Keywords as your comments as follows:
“……
Keywords: Piceatannol; Dendritic cells; Rheumatoid arthritis; inflammation; immunomodulation
……”
- Comments: L36-37-Please provide at least 1-2 specific examples for each type of these drugs, general categories are not enough.
Response: Dear reviewer, thank you to the reviewer for pointing out the insufficient description. We have replenished specific examples for each type of drugs as follows:
“……
Currently, various treatments for RA are widely utilized, including glucocorticoids (Dexamethasone and Prednisone) [5], NSAIDs (Aspirin and Celecoxib) [6], DMARDs (Iguratimod and Hydroxychloroquine) [7], as well as novel immunosuppressants (Methotrexate and Leflunomide) [8].
……”
- Comments: L37-Abbreviations, NSAIDs, DMARDs. Please check throughout the paper.
Response: Dear reviewer, thank you for pointing out our problems. we regret the error that occurred in the abbreviations of “NSAIDs and DMARDs”. We have conducted a comprehensive examination and adjustment of the manuscript.
- Comments: L45 - Punctuation error, “T、B lymphocytes”.
Response: Dear reviewer, thank you for pointing out our problems. This was a typographical error on our part. We have corrected the error of punctuation at L45 in the manuscript and have conducted a thorough review to ensure there are no other typographical errors.
- Comments: L77-78-This currently reads more like an abstract or conclusion. As an introduction, it should provide the background, purpose, potential outcomes, and broader implications of the research. General comment for Introduction - The structure of this part is somewhat chaotic: problems, existing solutions and shortcomings, problem targets, and then new solutions. The suggestion is to immediately follow the problem introduction with potential targets, and then highlight the shortcomings of existing drugs and introduce the advantages and characteristics of this study.
Response: Dear reviewer, thank you for your valuable feedback on our manuscript. We have rewrite the L77-78 following the structure as your comments and focusing on the immunosuppressive effect of PIC, because which have not been documented and it’s the the advantages and characteristics of our study. The modified content as follows:
“……
However, investigations into the other biological properties of PIC have not been documented, and research on whether it can treat rheumatoid arthritis by exerting an immunosuppressive effect to regulate immune balance remains unexplored.
To fill this gap, we identified the immunosuppressive effect of PIC through the DCs cell model induced by LPS in vivo and in vitro. We constructed the AIA mouse model uesing CFA and IFA to explore the immunomodulatory function of PIC in vivo. Network pharmacology, molecular docking and western blotting were employed to clarify the mechanisms by which PIC modulates immune balance. In general, our re-search aims to probe clearly the function and mechanisms of PIC targeting DCs in the treatment of rheumatoid arthritis.
……”
- Comments: L92-95-This is the results section not a discussion or conclusion, a brief summary is not enough. Please provide the specific numerical values of the indicator changes, as well as the p-value. Please check and revise this issue throughout the manuscript.
Response: Dear reviewer, thank you for pointing out our problems. We have replenished the p-value at L92-95 and revised this issue throughout the manuscript. In addition, due to the three biological replicates in each experimental group, we can’t provide the specific numerical values.
- Comments: L204-Formatting issue, “we explored...”.
Response: Dear reviewer, thank you to the reviewer for pointing out the insufficient description. We have revised it at L204 in manuscript as follows:
“……
The effect of PIC on the DCs-stimulated proliferation of OVA-specific T cells was explored in vitro.
……”
- Comments: L290-331-These contents overlap with the Introduction section. It is accepted to briefly mention the background and purpose of the study, but here is too long, too boring.
Response: Dear reviewer, we thank the reviewer for the constructive suggestion. We have revised the discussion of this part as follows:
“……
The pathogenesis of rheumatoid arthritis (RA) is extremely intricate, with pathogenic T cells playing an immediate role in the onset and development of the disease [29]. According to research reports, infiltration of CD4+ T cells at sites of inflammation is a defining characteristic of autoimmune diseases. In the early stages of RA, the synovial tissue of affected joints is populated by memory CD4+ T cells [30], which can further polarize into different phenotypes depending on the cytokine environment thereby participating in the immune response [31]. In addition, DCs are among the first cells to be activated in the synovial tissue of RA patients.
DCs act as a crucial bridge between the innate and adaptive immune systems, playing a vital role in maintaining immune tolerance within the body [32]. Immature DCs serve as sentinels of the immune system, primarily recognizing antigens through pattern recognition receptors and possessing a high phagocytic capacity. However, when stimulated by inflammatory mediators resulting from self-tissue damage, necrotic cells, or pathogens, immature DCs undergo maturation, which is characterized by elevated levels of MHC and co-stimulatory molecules [33,34]. This study found that under LPS-induced inflammatory conditions, PIC can downregulate the expression of surface molecules on DCs and reduce their secretion of inflammatory cytokines, thereby significantly inhibiting the maturation of LPS-induced DCs. Mature DCs, which express high levels of the chemokine receptor CCR-7, migrate to lymph nodes by interacting with CCL19 and CCL21, where they function in areas enriched with T cells [35,36]. Our findings indicate that PIC inhibits the migration of DCs from peripheral tissues to lymph nodes, consequently suppressing the activation and differentiation of T lymphocytes. Moreover, mature DCs possess a robust ability to pre-sent antigens; they process self-reactive antigens into peptides and present them to T cells while secreting IL-12, TNF-α, and TGF-β, which contribute to the activation of CD4+ T cells and their further differentiation into pro-inflammatory effector cells in-volved in inflammatory responses and joint damage [37,38]. However, the activation and differentiation of CD4+ T cells were inhibited by DCs treated with PIC. Further-more, the immunosuppressive effect of PIC was also identified in AIA mouse model, where it exerted similar effects on the maturation and function of DCs and the activation and differentiation of CD4+ T cells as observed in vitro.
……”
- Comments: L332-“The current study”, for yours or others? I suppose that here should be “this study”or “the present study”.
Response: Dear reviewer, thank you to the reviewer for pointing out the insufficient description. “The current study” in L332 means “this study”, and we have revised this issue in manuscript.
- Comments: L332-361-These contents overlap with the Result section, all of which are brief conclusions without any discussion. The discussion section should showcase how the authors derived these conclusions from data and image results.
Response: Dear reviewer, thank you for pointing out our problems. We have revised these contents based on recently literatures and the results as follows:
“……
DCs act as a crucial bridge between the innate and adaptive immune systems, playing a vital role in maintaining immune tolerance within the body [32]. Immature DCs serve as sentinels of the immune system, primarily recognizing antigens through pat-tern recognition receptors and possessing a high phagocytic capacity. However, when stimulated by inflammatory mediators resulting from self-tissue damage, necrotic cells, or pathogens, immature DCs undergo maturation, which is characterized by elevated levels of MHC and co-stimulatory molecules [33,34]. This study found that under LPS-induced inflammatory conditions, PIC can downregulate the expression of surface molecules on DCs and reduce their secretion of inflammatory cytokines, thereby significantly inhibiting the maturation of LPS-induced DCs. Mature DCs, which express high levels of the chemokine receptor CCR-7, migrate to lymph nodes by interacting with CCL19 and CCL21, where they function in areas enriched with T cells [35,36]. Our findings indicate that PIC inhibits the migration of DCs from peripheral tissues to lymph nodes, consequently suppressing the activation and differentiation of T lymphocytes. Moreover, mature DCs possess a robust ability to present antigens; they process self-reactive antigens into peptides and present them to T cells while secreting IL-12, TNF-α, and TGF-β, which contribute to the activation of CD4+ T cells and their further differentiation into pro-inflammatory effector cells involved in inflammatory responses and joint damage [37,38]. However, the activation and differentiation of CD4+ T cells were inhibited by DCs treated with PIC. Furthermore, the immunosuppressive effect of PIC was also identified in AIA mouse model, where it exerted similar effects on the maturation and function of DCs and the activation and differentiation of CD4+ T cells as observed in vitro.
PIC, as a hydroxylated derivative of resveratrol, is derived from various plants [39]. Like resveratrol, PIC possess potent antioxidant activity, as well as an-ti-inflammatory and anticancer properties [40]. This study demonstrates that PIC not only inhibits the maturation of DCs, but also suppresses T cell activation. However, both resveratrol and PIC were found to be cytotoxic to macrophages in the MTT viability test, although this cytotoxicity was significantly mitigated co-treatment with zymosan [41]. Additionally, PIC has been shown to ameliorate inflammation by the inhibiting TLR4/NF-κB/NLRP3 pathway in hepatic macrophages [42]. Furthermore, Piceatannol is more effective than resveratrol in suppressing adipogenesis in human visceral adipose-derived stem cells due to its distinct antilipolytic properties [43-44].
In addition, PIC exhibits diverse physiological activities through its multi-pathway and multi-target properties. In our study, we found that PIC interacts with targets related to DCs maturation and function. Specifically, PIC interacts with MAPK14 and RELA, thereby influencing the NF-κB and MAPK signaling pathways, which inhibit the immune activation function of DCs, thus achieving the purpose of treating autoimmune arthritis. Furthermore, MTX, a potent competitive inhibitor of dihydrofolate reductase, has been shown to possess potential organ toxicity. Our findings indicate that the serum levels of pro-inflammatory cytokines after PIC treatment were lower than those in the MTX group, and PIC was safer than MTX [45-46].
……”
- Comments: L365-367-Please provide the approval code and validity period.
Response: Dear reviewer, thank you for pointing out our problems. We apologize for neglecting the approval code and validity period and we have replenished it in manuscript as follows:
“……
Female BALB/c mice aged 6-8 weeks were purchased from the Experimental Animal Center of Xinjiang Medical University. Animal experiment was conducted under the guidance of the Animal Care and Use Committee of the College of Life Science and Technology, Xinjiang University (XJUAE-2024-041).
……”
- Comments: L518-Abbreviations used in the manuscript are not fully provided.
Response: Dear reviewer, thank you for pointing out our problems. We sincerely apologize for our carelessness. We have replenished the abbreviations in table as follows:
“……
PIC |
Piceatannol |
RA |
Rheumatoid arthritis |
DCs |
Dendritic cells |
LC-MS APC CCR-7 CIA DMARDs NSAIDs TCRs MAPK NF-κB LPS CFSE RBC Lysate OVA antigen BCA kit CFA MTX Treg EGCG C3G AIA PPI GO KEGG |
Liquid Chromatography-Mass Spectrometry Antigen presenting cell CC chemokine receptor 7 CFA-induced arthritis Disease-modifying anti-rheumatic drugs Non-steroidal anti-inflammatory drugs T cell receptors Mitogen-activated protein kinase Nuclear Factor kappa-B Lipopolysaccharide 5,6-CarboxyfluoresceinDiacetate Succinimidyl Ester Red blood cell lysate Ovalbumin antigen Bicinchoninic Acid Kit Complete Freund's adjuvant Methotrexate Regulatory T cells Epigallocatechin gallate Cyanidin-3-O-glucoside adjuvant-induced arthritis protein-protein interaction Gene Ontology Kyoto Encyclopedia of Genes and Genomes |
……”
We tried our best to improve the manuscript and we appreciate for Editors/Reviewer’s warm work earnestly, and hope that the correction will meet with approval.
Once again, thank you very much for your comments and suggestions.
Best wishes.

Reviewer 2 Report
Comments and Suggestions for Authors
The manuscript explores the immunomodulatory effects of Piceatannol (PIC) on dendritic cells (DCs) and its therapeutic potential in experimental arthritis. It provides a comprehensive examination of PIC's impact on immune cell function, detailing its mechanism through MAP kinase and NF-κB signaling pathways. The combination of in vitro, in vivo, and network pharmacology approaches strengthens the study’s scientific rigor.
Major Concerns & Suggestions
- Clarity & Readability
While the manuscript presents valuable findings, the readability can be significantly improved. Some sections contain awkward phrasing, grammatical errors, and redundant sentences.
Examples:“Recently, natural small molecules have been developed as alternative treating substance.”Revision: “Recently, natural small molecules have been explored as alternative therapeutic agents.”
-"This PIC-skewed Th differentiation profile was corelated with that of corresponding polarizing cytokine secretion from DCs in the CIA model." Revised:"The PIC-induced shift in Th differentiation correlated with the secretion of polarizing cytokines from DCs in the CIA model."
- "Therefore, the development of effective therapeutic drugs with low toxicity and minimal side effects remains a top priority." This sentence is generic. The authors should connect it directly to PIC, emphasizing its potential advantages over existing treatments.
Recommendation: The manuscript should undergo thorough English language editing to enhance clarity, conciseness, and fluency.
- Justification of PIC Dosage & Toxicity Studies
The study mentions that 2 µM PIC is the optimal dose for inhibiting DC maturation without affecting cell viability.
- How was this concentration selected?
The authors should provide more data on the dose-response relationship. Were lower doses ineffective? Were higher doses toxic? - Toxicity assessments in vivo?
While liver and kidney function tests were performed, a histological analysis of major organs (e.g., liver, kidney, spleen) could provide stronger evidence of safety.
- Mechanistic Analysis – Limited Discussion on Potential Off-Target Effects
The study focuses primarily on MAPK and NF-κB pathways.
- Are there any off-target effects?
- Could PIC impact other immune cell types, such as macrophages or neutrophils?
- Could PIC affect bone metabolism, given its role in arthritis?
- A brief discussion on possible unintended effects of PIC would improve the completeness of the study.
- Figures & Statistical Clarity
Figure 1 (Flow Cytometry Data)
- The gating strategy for flow cytometry should be described more clearly.
- The axes and fluorescence markers should be clearly labeled.
Figure 6 (Network Pharmacology & Molecular Docking)
- The binding affinity scores should be directly compared with known inhibitors to validate the strength of interactions.
- Include detailed legends explaining the statistical comparisons (e.g., was a t-test or ANOVA used?).
- Ensure that figures are high resolution for clarity in publication.
- Literature Contextualization
The discussion primarily focuses on PIC, but lacks a broader comparison with other small-molecule immunosuppressants.
Suggested Comparisons:
- Resveratrol: Since PIC is a derivative of resveratrol, the manuscript should include a comparison of their effects on DCs and T cells.
- Other RA treatments: How does PIC compare to methotrexate (MTX) or JAK inhibitors?
- English Language Quality & Grammar Issues
The manuscript contains several grammatical errors, typos, and awkward sentence structures.
Examples & Suggested Revisions:
- “In current study, we validated that PIC can inhibit DCs maturation and cytokine secretion…”
Revision:
“In this study, we confirmed that PIC inhibits DC maturation and cytokine secretion…” - “With the maturation, MHC I and II molecules and co-stimulatory molecules are highly expressed on DCs surface…”
Revision: “During maturation, MHC I/II and co-stimulatory molecules are highly expressed on the surface of DCs…”
The manuscript contains several grammatical errors, typos, and awkward sentence structures.
Author Response
Subject: Letter of major revision
Manuscript ID: ijms-3473583
Title: Piceatannol inhibits the immunostimulatory functions of dendritic cells and alleviates experimental arthritis
Dear Reviewers:
We thank you very much for your patient review and important suggestions for our article. Those comments are all valuable and very helpful for revising and improving our paper, as well as the important guiding significance to our researches. We are very sorry to update the revised manuscript so late, because the adding of some necessary information. We have improved the parts that you mentioned.
In this revised version, we have addressed the concerns of the reviewers. An item-by-item response to the reviewer’s comments is enclosed, and the revision was marked in red fronts in the manuscript. We hope that these revisions successfully address concerns and requirements and that this manuscript will be accepted.
The main corrections in the paper and the responds to the reviewers’ comments are as following:
Reviewer #2
- Comment: Clarity & Readability
- While the manuscript presents valuable findings, the readability can be significantly improved. Some sections contain awkward phrasing, grammatical errors, and redundant sentences. Examples:“Recently, natural small molecules have been developed as alternative treating substance.”Revision:“Recently, natural small molecules have been explored as alternative therapeutic agents.” "This PIC-skewed Th differentiation profile was corelated with that of corresponding polarizing cytokine secretion from DCs in the CIA model." Revised:"The PIC-induced shift in Th differentiation correlated with the secretion of polarizing cytokines from DCs in the CIA model." "Therefore, the development of effective therapeutic drugs with low toxicity and minimal side effects remains a top priority." This sentence is generic. The authors should connect it directly to PIC, emphasizing its potential advantages over existing treatments.
- Recommendation: The manuscript should undergo thorough English language editingto enhance clarity, conciseness, and fluency.
Response: Dear reviewer, thank you very much for your comprehensive review and for pointing out the issue. We have carefully considered your feedback and revised it in manuscript, we further made the corrections to ensure consistency throughout the whole manuscript.
- Comments: Justification of PIC Dosage & Toxicity Studies.
- The study mentions that 2 µM PIC is the optimal dose for inhibiting DC maturation without affecting cell viability. How was this concentration selected? The authors should provide more data on the dose-response relationship. Were lower doses ineffective? Were higher doses toxic?
Response: Dear reviewer, thank you for your thorough review of our paper and your valuable feedback. In our research, the initial dose of PIC was determined by reviewing the literature [1], and then we investigated the safe dose of PIC on DCs (by measuring cell viability) in a gradually decreasing dose. Additionally, we investigated the effective dose of PIC through the DCs cell model induced by LPS-induced with PIC. Finally, we found that 2 µM PIC is the optimal dose for inhibiting DCs maturation without affecting cell viability, but when the concentration is greater than 2 µM, it could affect cell viability. Low dose has no toxicity but poor effect on DC maturation.
- Toxicity assessments in vivo? While liver and kidney function tests were performed, a histological analysis of major organs (e.g., liver, kidney, spleen) could provide stronger evidence of safety.
Response: Dear reviewer, thank you very much for your thorough review and for pointing out the issue. It was an oversight that we have not performed the histological analysis of major organs. However, we provided the organ index of major organs (e.g., liver, kidney, spleen) in supplementary results, which proved the safety of PIC in vivo. In addition, we will be carrying experiments to explore the histological analysis of major organs in the later experiment.
- Comments: Mechanistic Analysis–Limited Discussion on Potential Off-Target Effects.
- The study focuses primarily on MAPK and NF-κB pathways.
- Are there any off-target effects?
- Could PIC impact other immune cell types, such as macrophages or neutrophils?
- The study focuses primarily on MAPK and NF-κB pathways.
- Could PIC affect bone metabolism, given its role in arthritis?
- A brief discussion on possible unintended effects of PIC would improve the completeness of the study.
Response: Dear reviewer, thank you very much for your thorough review and for pointing out the issue. It’s regret that we neglect the off-target effects of PIC, which will be taken into account in the next experiment. In addition, based on the role of DCs in activating T cells in RA, we only focused on the impact of PIC on DCs in vitro and vivo, it’s our cursoriness to explore the impact of PIC on other immune cell types, we will take it into consideration in the next experiment. In our results, we found that PIC alleviated the histopathological changing of joint tissues (Figure 3D), but we haven’t detected the bone metabolism, which needs to be tested for a long time. We will take this into account in our later experiments. Thank you very much for your constructive suggestion again, we have replenished a brief discussion about the potential off-target effects and possible unintended effects of PIC in manuscript as follows:
“……
PIC, as a hydroxylated derivative of resveratrol, is derived from various plants [39]. Like resveratrol, PIC possess potent antioxidant activity, as well as anti-inflammatory and anticancer properties [40]. This study demonstrates that PIC not only inhibits the maturation of DCs, but also suppresses T cell activation. However, both resveratrol and PIC were found to be cytotoxic to macrophages in the MTT viability test, although this cytotoxicity was significantly mitigated co-treatment with zymosan [41]. Additionally, PIC has been shown to ameliorate inflammation by the inhibiting TLR4/NF-κB/NLRP3 pathway in hepatic macrophages [42]. Furthermore, Piceatannol is more effective than resveratrol in suppressing adipogenesis in human visceral adipose-derived stem cells due to its distinct antilipolytic properties [43-44].
In addition, PIC exhibits diverse physiological activities through its multi-pathway and multi-target properties. In our study, we found that PIC interacts with targets related to DCs maturation and function. Specifically, PIC interacts with MAPK14 and RELA, thereby influencing the NF-κB and MAPK signaling pathways, which inhibit the immune activation function of DCs, thus achieving the purpose of treating autoimmune arthritis. Furthermore, MTX, a potent competitive inhibitor of dihydrofolate reductase, has been shown to possess potential organ toxicity. Our findings indicate that the serum levels of pro-inflammatory cytokines after PIC treatment were lower than those in the MTX group, and PIC was safer than MTX [45-46].
……”
- Comments: Figures & Statistical Clarity.
- Figure 1 (Flow Cytometry Data)
- The gating strategy for flow cytometry should be described more clearly.
- The axes and fluorescence markers should be clearly labeled.
- Figure 1 (Flow Cytometry Data)
Response: Dear reviewer, thank you very much for your valuable reminder. We have replenished the gating strategy for flow cytometry, and labeled the axes and fluorescence markers in Figure 1 in manuscript.
- Figure 6 (Network Pharmacology & Molecular Docking)
- The binding affinity scores should be directly compared with known inhibitors to validate the strength of interactions.
Response: Dear reviewer, thank you very much for your constructive suggestion. We have replenished the molecular docking results of known inhibitors in manuscript.
- Include detailed legends explaining the statistical comparisons (e.g., was a t-test or ANOVA used?).
- Ensure that figures are high resolution for clarity in publication.
Response: Dear reviewer, thank you very much for your valuable reminder. In this section, statistical significance was determined using one-way analysis of variance (ANOVA) or unpaired t-test, with P < 0.05 considered statistically significant, which was described at the section of 4.11. In addition, we have replaced the figures in the manuscript with high-resolution versions to ensure clarity in publication.
- Comments: Literature Contextualization.
- The discussion primarily focuses on PIC, but lacks a broader comparison with other small-molecule immunosuppressants. Suggested Comparisons:
- Resveratrol: Since PIC is a derivative of resveratrol, the manuscript should include a comparison of their effects on DCs and T cells.
- Other RA treatments: How does PIC compare to methotrexate (MTX) or JAK inhibitors?
- The discussion primarily focuses on PIC, but lacks a broader comparison with other small-molecule immunosuppressants. Suggested Comparisons:
Response: Dear reviewer, thank you very much for your constructive suggestions. We have replenished the comparison between PIC and Resveratrol, PIC and methotrexate in discussion as follows:
“……
PIC, as a hydroxylated derivative of resveratrol, is derived from various plants [39]. Like resveratrol, PIC possess potent antioxidant activity, as well as anti-inflammatory and anticancer properties [40]. This study demonstrates that PIC not only inhibits the maturation of DCs, but also suppresses T cell activation. However, both resveratrol and PIC were found to be cytotoxic to macrophages in the MTT viability test, although this cytotoxicity was significantly mitigated co-treatment with zymosan [41]. Additionally, PIC has been shown to ameliorate inflammation by the inhibiting TLR4/NF-κB/NLRP3 pathway in hepatic macrophages [42]. Furthermore, Piceatannol is more effective than resveratrol in suppressing adipogenesis in human visceral adipose-derived stem cells due to its distinct antilipolytic properties [43-44].
In addition, PIC exhibits diverse physiological activities through its multi-pathway and multi-target properties. In our study, we found that PIC interacts with targets related to DCs maturation and function. Specifically, PIC interacts with MAPK14 and RELA, thereby influencing the NF-κB and MAPK signaling pathways, which inhibit the immune activation function of DCs, thus achieving the purpose of treating autoimmune arthritis. Furthermore, MTX, a potent competitive inhibitor of dihydrofolate reductase, has been shown to possess potential organ toxicity. Our findings indicate that the serum levels of pro-inflammatory cytokines after PIC treatment were lower than those in the MTX group, and PIC was safer than MTX [45-46].
……”
- Comments: English Language Quality & Grammar Issues
- The manuscript contains several grammatical errors, typos, and awkward sentence structures. Examples & Suggested Revisions:
- “In current study, we validated that PIC can inhibit DCs maturation and cytokine secretion…”
- The manuscript contains several grammatical errors, typos, and awkward sentence structures. Examples & Suggested Revisions:
Revision: “In this study, we confirmed that PIC inhibits DC maturation and cytokine secretion…”
- “With the maturation, MHC I and II molecules and co-stimulatory molecules are highly expressed on DCs surface…”
Revision: “During maturation, MHC I/II and co-stimulatory molecules are highly expressed on the surface of DCs…”
Response: Dear reviewer, thank you very much for your constructive suggestions. We have revised the grammatical errors, typos, and awkward sentence structures as you mentioned to ensure consistency throughout the whole manuscript. Thank you for pointing out these errors.
Reference
- Gao, X.; Kang, X.; Lu, H.; Xue, E.; Chen, R.; Pan, J.; Ma, J. Piceatannol suppresses inflammation and promotes apoptosis in rheumatoid arthritis‑fibroblast‑like synoviocytes by inhibiting the NF-κB and MAPK signaling pathways. Mol. Med. Rep. 2022, 25(5), 180. https://doi.org/10.3892/mmr.2022.12696
We tried our best to improve the manuscript and we appreciate for Editors/Reviewer’s warm work earnestly, and hope that the correction will meet with approval.
Once again, thank you very much for your comments and suggestions.
Best wishes.

Reviewer 3 Report
Comments and Suggestions for Authors
The manuscript submitted to IJMS presents a quality research work where the introduction and its background, the methodology and the presentation of the results are remarkable. Therefore, I consider that it has a place in this high-impact journal.
However, I consider that there are certain aspects that should be improved:
1. The figures presented in the results cannot be seen. The authors should present them in another format where these figures have a larger size.
2. I consider that the Discussion is quite poor considering the volume of results obtained. I recommend that the authors include several sections for the Discussion, as well as the results. In this way, I consider that the work will achieve a higher quality.
3. Include a Conclusions section.
Author Response
Subject: Letter of major revision
Manuscript ID: ijms-3473583
Title: Piceatannol inhibits the immunostimulatory functions of dendritic cells and alleviates experimental arthritis
Dear Reviewers:
We thank you very much for your patient review and important suggestions for our article. Those comments are all valuable and very helpful for revising and improving our paper, as well as the important guiding significance to our researches. We are very sorry to update the revised manuscript so late, because the adding of some necessary information. We have improved the parts that you mentioned.
In this revised version, we have addressed the concerns of the reviewers. An item-by-item response to the reviewer’s comments is enclosed, and the revision was marked in red fronts in the manuscript. We hope that these revisions successfully address concerns and requirements and that this manuscript will be accepted.
The main corrections in the paper and the responds to the reviewers’ comments are as following:
Reviewer #3
- Comment: The figures presented in the results cannot be seen. The authors should present them in another format where these figures have a larger size.
Response: Dear reviewer, thank you very much for your notification. We have updated these figures in another format in manuscript.
- Comments: I consider that the Discussion is quite poor considering the volume of results obtained. I recommend that the authors include several sections for the Discussion, as well as the results. In this way, I consider that the work will achieve a higher quality.
Response: Dear reviewer, thank you for your valuable feedback. We appreciate and have accepted your suggestion. We have revised the Discussion as your comments as follows:
“……
- Discussion
The pathogenesis of rheumatoid arthritis (RA) is extremely intricate, with pathogenic T cells playing an immediate role in the onset and development of the disease [29]. According to research reports, infiltration of CD4+ T cells at sites of inflammation is a defining characteristic of autoimmune diseases. In the early stages of RA, the synovial tissue of affected joints is populated by memory CD4+ T cells [30], which can further polarize into different phenotypes depending on the cytokine environment thereby participating in the immune response [31]. In addition, DCs are among the first cells to be activated in the synovial tissue of RA patients.
DCs act as a crucial bridge between the innate and adaptive immune systems, playing a vital role in maintaining immune tolerance within the body [32]. Immature DCs serve as sentinels of the immune system, primarily recognizing antigens through pattern recognition receptors and possessing a high phagocytic capacity. However, when stimulated by inflammatory mediators resulting from self-tissue damage, necrotic cells, or pathogens, immature DCs undergo maturation, which is characterized by elevated levels of MHC and co-stimulatory molecules [33,34]. This study found that under LPS-induced inflammatory conditions, PIC can downregulate the expression of surface molecules on DCs and reduce their secretion of inflammatory cytokines, thereby significantly inhibiting the maturation of LPS-induced DCs. Mature DCs, which express high levels of the chemokine receptor CCR-7, migrate to lymph nodes by interacting with CCL19 and CCL21, where they function in areas enriched with T cells [35,36]. Our findings indicate that PIC inhibits the migration of DCs from peripheral tissues to lymph nodes, consequently suppressing the activation and differentiation of T lymphocytes. Moreover, mature DCs possess a robust ability to pre-sent antigens; they process self-reactive antigens into peptides and present them to T cells while secreting IL-12, TNF-α, and TGF-β, which contribute to the activation of CD4+ T cells and their further differentiation into pro-inflammatory effector cells in-volved in inflammatory responses and joint damage [37,38]. However, the activation and differentiation of CD4+ T cells were inhibited by DCs treated with PIC. Further-more, the immunosuppressive effect of PIC was also identified in AIA mouse model, where it exerted similar effects on the maturation and function of DCs and the activation and differentiation of CD4+ T cells as observed in vitro.
PIC, as a hydroxylated derivative of resveratrol, is derived from various plants [39]. Like resveratrol, PIC possess potent antioxidant activity, as well as an-ti-inflammatory and anticancer properties [40]. This study demonstrates that PIC not only inhibits the maturation of DCs, but also suppresses T cell activation. However, both resveratrol and PIC were found to be cytotoxic to macrophages in the MTT viability test, although this cytotoxicity was significantly mitigated co-treatment with zymosan [41]. Additionally, PIC has been shown to ameliorate inflammation by the inhibiting TLR4/NF-κB/NLRP3 pathway in hepatic macrophages [42]. Furthermore, Piceatannol is more effective than resveratrol in suppressing adipogenesis in human visceral adipose-derived stem cells due to its distinct antilipolytic properties [43-44].
In addition, PIC exhibits diverse physiological activities through its multi-pathway and multi-target properties. In our study, we found that PIC interacts with targets related to DCs maturation and function. Specifically, PIC interacts with MAPK14 and RELA, thereby influencing the NF-κB and MAPK signaling pathways, which inhibit the immune activation function of DCs, thus achieving the purpose of treating autoimmune arthritis. Furthermore, MTX, a potent competitive inhibitor of dihydrofolate reductase, has been shown to possess potential organ toxicity. Our findings indicate that the serum levels of pro-inflammatory cytokines after PIC treatment were lower than those in the MTX group, and PIC was safer than MTX [45-46].
.……”
- Comments: Include a Conclusions section.
Response: Dear reviewer, we thank the reviewer for the constructive suggestion. We have replenished a conclusion section under the section of Discussion as follows:
“……
Conclusion
In general, we demonstrated that PIC exhibits an anti-inflammatory function in the AIA mouse model by suppressing the maturation and secretion of pro-inflammatory cytokines from DCs in the lymph nodes and spleen, further re-straining the differentiation of Th17 and Th1 cells in spleen, so as to decreased the levels of IL-6, TNF-α and IL-17A and so on in peripheral blood. The combined analysis of network pharmacology and in vitro experiments further proved the immunosuppressive effect of PIC by suppressing the maturation and immune activation function of DCs via the MAPK and NF-κB signaling pathways, which was further validated by molecular docking and Western blotting. Our research proved the immunoregulating function of PIC on the maturation and function of DCs and AIA mouse model, and PIC may exert its application in the treatment of RA patients.
.……”
We tried our best to improve the manuscript and we appreciate for Editors/Reviewer’s warm work earnestly, and hope that the correction will meet with approval.
Once again, thank you very much for your comments and suggestions.
Best wishes.

Round 2
Reviewer 1 Report
Comments and Suggestions for Authors
The authors have adequately addressed the majority of my previous comments and suggestions, resulting in a significantly improved manuscript. I have no further comments at this time and, in my opinion, the current version is acceptable for publication.
Author Response
Manuscript ID: ijms-3473583
Title: Piceatannol inhibits the immunostimulatory functions of dendritic cells and alleviates experimental arthritis
Dear Reviewers:
We thank you very much for your patient review and important suggestions for our article.
Best wishes.
Reviewer 2 Report
Comments and Suggestions for Authors
The authors have undertaken a comprehensive revision of the manuscript in response to the first round of peer review. The revised version shows significant improvement in clarity, data presentation, mechanistic interpretation, and contextualization of findings. The effort made to address major and minor issues is appreciated and has strengthened the scientific rigor and impact of the study.
However, while most of the initial concerns have been adequately addressed, a few areas would benefit from additional clarification or minor revision before the manuscript can be considered for final acceptance.
Areas Needing Improvement
- Language and Clarity
- While the authors have made commendable efforts to improve grammar and flow, several sentences still contain awkward phrasing, redundant wording, and grammatical inconsistencies. Examples: “With the maturation, MHC I and II molecules…” should be revised to: “During maturation, MHC I and II molecules…”
- “We validated that PIC can inhibit DCs maturation…” → “We confirmed that PIC inhibits DC maturation…” A final round of comprehensive English editing is essential to ensure fluency and clarity throughout.
- Mechanistic Off-Target Discussion
- The manuscript now includes a brief discussion on potential off-target effects of PIC. However, this section would benefit from further detail and structure.
- Currently, the discussion mentions effects on macrophages and adipose tissue, but does not sufficiently explore possible systemic immunomodulatory risks, especially in long-term use. Add a few sentences explicitly discussing the broader immunological impact of PIC on other immune cell populations (e.g., neutrophils, B cells), and whether long-term modulation of DCs might impair host defense or tolerance.
- Histological Assessment of Toxicity
- The authors provided organ index data and acknowledge the lack of histological validation.
- While this is acceptable at the current stage, a stronger justification for omitting histopathology would improve transparency. Include a sentence in the discussion explaining why histological assessment was not feasible and clarify plans for inclusion in future studies.
Comments on the Quality of English LanguageThe overall quality of English in the revised manuscript has improved compared to the original submission. However, several grammatical errors, awkward sentence structures, and non-standard phrasing remain throughout the text. These issues occasionally affect the clarity and flow of scientific communication.
To ensure the manuscript meets publication standards and is understandable to an international audience, a final round of thorough English language is strongly recommended. Attention should be paid to verb tenses, article usage, subject-verb agreement, and sentence conciseness.
Author Response
Subject: Letter of minor revisions
Manuscript ID: ijms-3473583
Title: Piceatannol inhibits the immunostimulatory functions of dendritic cells and alleviates experimental arthritis
Dear Reviewers:
We thank you very much for your patient review and important suggestions for our article. Those comments are all valuable and very helpful for revising and improving our paper, as well as the important guiding significance to our researches. We are very sorry to update the revised manuscript so late, because the adding of some necessary information. We have improved the parts that you mentioned.
In this revised version, we have addressed the concerns of the reviewers. An item-by-item response to the reviewer’s comments is enclosed, and the revision was marked in red fronts in the manuscript. We hope that these revisions successfully address concerns and requirements and that this manuscript will be accepted.
The main corrections in the paper and the responds to the reviewers’ comments are as following:
Reviewer #2
- Comment:Language and Clarity
- While the authors have made commendable efforts to improve grammar and flow, several sentences still contain awkward phrasing, redundant wording, and grammatical inconsistencies. Examples: “With the maturation, MHC I and II molecules…” should be revised to: “During maturation, MHC I and II molecules…”. “We validated that PIC can inhibit DCs maturation…” → “We confirmed that PIC inhibits DC maturation…” A final round of comprehensive English editing is essential to ensure fluency and clarity throughout.
Response: Dear reviewer, thank you very much for your comprehensive review and for pointing out the issue. We have carefully considered your feedback and revised it in manuscript. We further made the corrections to ensure consistency throughout the whole manuscript. Thank you for pointing out these errors.
- Comments: Mechanistic Off-Target Discussion.
- The manuscript now includes a brief discussion on potential off-target effects of PIC. However, this section would benefit from further detail and structure.
- Currently, the discussion mentions effects on macrophages and adipose tissue, but does not sufficiently explore possible systemic immunomodulatory risks, especially in long-term use. Add a few sentences explicitly discussing the broader immunological impact of PIC on other immune cell populations (e.g., neutrophils, B cells), and whether long-term modulation of DCs might impair host defense or tolerance.
Response: Dear reviewer, Thank you very much for your detailed inquiries. We have replenished the broader immunological impact of PIC on other immune cell populations (L359-360). And the potential impact of long-term modulation of DC on host defense or tolerance will be explored in the later experiment. We replenished a sentence about the long-term effect of DC in discussion (L374-378) as follows:
“……
PIC possesses the ability of reducing the toxic action of neutrophils through the involve-ment of protein kinase C [45].
Methotrexate (MTX) is currently the first-line treatment for RA; however, prolonged use may lead to a range of adverse effects [46-47]. Our results indicated that the serum levels of pro-inflammatory cytokines after PIC treatment were lower than those in the MTX group. In addition, PIC exhibited diverse physiological activities through its multi-pathway and multi-target properties. In this study, we used network pharmacology to predict that PIC could interact with targets related to DCs maturation and function. PIC interacted with MAPK14 and RELA to regulate the activation of NF-κB and MAPK signaling path-ways in DCs by molecular docking and Western blot. PIC can enhance therapeutic efficacy by inhibiting DCs maturation via multiple targets. However, given that the MAPK path-way is implicated in numerous cellular life processes, the inhibition of this pathway by PIC may lead to potential off-target effects. Therefore, further investigations are warranted to comprehensively assess the safety profile of PIC.
Of course, our study has several limitations that require further investigation. On the one hand, In AIA mice model, the duration was short, and the changes of immune homeostasis in mice after long-term treatment not be observed. On the other hand, a systematic assessment of PIC safety is lacking, which will be further investigated in future studies.
……”
- Comments: Histological Assessment of Toxicity.
- The authors provided organ index data and acknowledge the lack of histological validation.While this is acceptable at the current stage, a stronger justification for omitting histopathology would improve transparency. Include a sentence in the discussion explaining why histological assessment was not feasible and clarify plans for inclusion in future studies.
Response: Dear reviewer, thank you very much for your thorough review and for pointing out the issue. We replenished a sentence about the histological assessment and plans for inclusion in future studies as follows:
“……
Of course, our study has several limitations that require further investigation. On the one hand, In AIA mice model, the duration was short, and the changes of immune homeostasis in mice after long-term treatment not be observed. On the other hand, a systematic assessment of PIC safety is lacking, which will be further investigated in future studies.
……”
- Comments: The overall quality of English in the revised manuscript has improved compared to the original submission. However, several grammatical errors, awkward sentence structures, and non-standard phrasing remain throughout the text. These issues occasionally affect the clarity and flow of scientific communication.
To ensure the manuscript meets publication standards and is understandable to an international audience, a final round of thorough English language is strongly recommended. Attention should be paid to verb tenses, article usage, subject-verb agreement, and sentence conciseness.
Response: Dear reviewer, thank you very much for your constructive suggestions. We have revised the grammatical errors, typos, and awkward sentence structures as you mentioned to ensure consistency throughout the whole manuscript. Thank you for pointing out these errors.
We tried our best to improve the manuscript and we appreciate for Editors/Reviewer’s warm work earnestly, and hope that the correction will meet with approval.
Once again, thank you very much for your comments and suggestions.
Best wishes.

Reviewer 3 Report
Comments and Suggestions for Authors
The manuscript has been significantly improved by the authors, following my instructions and those of the other colleagues who reviewed the work.
The figures are more visible and have significantly improved their appearance.
The authors also now provide a Conclusions section, necessary to address the objective of the work.
The Discussion section has also been improved; however, my suggestion to create several subsections was not taken into account. Despite this, the changes can be considered sufficient here.
For all these reasons, I consider the manuscript suitable for acceptance without modifications.
Author Response

(The authors gave the same response as above.)
